# Development of a Hydrophobicity-Controlled Delivery System Containing Levodopa Methyl Ester Hydrochloride Loaded into a Mesoporous Silica

**DOI:** 10.3390/pharmaceutics13071039

**Published:** 2021-07-07

**Authors:** Tamás Kiss, Gábor Katona, László Mérai, László Janovák, Ágota Deák, Gábor Kozma, Zoltán Kónya, Rita Ambrus

**Affiliations:** 1Institute of Pharmaceutical Technology and Regulatory Affairs, Faculty of Pharmacy, University of Szeged, H-6720 Szeged, Hungary; tamas.kiss@pharm.u-szeged.hu (T.K.); katona.gabor@szte.hu (G.K.); 2Department of Physical Chemistry and Materials Science, Faculty of Science and Informatics, University of Szeged, H-6720 Szeged, Hungary; merail@chem.u-szeged.hu (L.M.); janovakl@chem.u-szeged.hu (L.J.); deak.agota@stud.u-szeged.hu (Á.D.); 3Department of Applied and Environmental Chemistry, Faculty of Science and Informatics, University of Szeged, H-6720 Szeged, Hungary; kozmag@chem.u-szeged.hu (G.K.); konya@chem.u-szeged.hu (Z.K.)

**Keywords:** mesoporous silica, levodopa methyl ester hydrochloride, melevodopa hydrochloride, initial drug release, physical stability, hydrophobization, sylilation, Syloid, surface area, incorporation

## Abstract

Background: The drug release of antiparkinsonian drugs is an important issue during the formulation process because proper release kinetics can help to reduce the off periods of Parkinson’s disease. A 2-factor, 3-level (3^2^) full-factorial design was conducted to evaluate statistically the influence of the hydrophobicity of mesoporous silica on drug release. Methods: Hydrophobization was evaluated by different methods, such as contact angle measurement, infrared spectroscopy and charge titration. After loading the drug (levodopa methyl ester hydrochloride, melevodopa hydrochloride, LDME) into the mesopores, drug content, particle size, specific surface area and homogeneity of the products were also analyzed. The amorphous state of LDME was verified by X-ray diffractometry and differential scanning calorimetry. Results: Drug release was characterized by a model-independent method using the so-called initial release rate parameter, as detailed in the article. The adaptability of this method was verified; the model fitted closely to the actual release results according to the similarity factor, independently of the release kinetics. Conclusions: The API was successfully loaded into the silica, resulting in a reduced surface area. The release studies indicated that the release rate significantly decreased (*p* < 0.05) with increasing hydrophobicity. The products with controlled release can reduce the off period frequency.

## 1. Introduction

Several active pharmaceutical ingredients (APIs) have a narrow therapeutic index, such as levodopa (LD). In the case of these APIs, there is an increased need for appropriate release control [1].

Nowadays, Parkinson’s disease is mostly treated by LD [2,3]. As the traditional—orally administered—LD therapy progresses, the therapeutic window (on time) becomes narrow, and the blood level becomes unpredictable [4]. This leads to side effects, called LD-related motor complications (LRMCs) [5,6], resulting in the so-called on-off phenomenon [7,8], which could be managed by the efficient control of drug release.

A limitation of LD formulations is the low solubility of the API, which leads to poorer release control; therefore, intensive research was carried out to find more soluble derivatives of LD in the last decades [9]. Ester prodrugs seem promising, among which levodopa methyl ester (melevodopa) is the most studied [9]. Melevodopa is much more soluble than LD, and its absorption ability is higher thanks to its higher lipophilicity [10]. Its hydrochloride salt is marketed as an effervescent tablet under the name of Sirio in Italy (Chiesi Farmaceutici SpA, Parma, Italy) [11].

Surface-modified mesoporous silicas (MPSs) might be promising excipient candidates for the release control of the antiparkinsonian API. MPS microparticles are 3D nanostructures because the pore diameter falls within the 2–50 nm range. The high pore volume and specific surface area of these materials ensure the adsorption of the drug molecules in the required amount. The dominant functional groups of the silica surfaces are siloxane (Si-O-Si) and silanol (-Si-OH) groups, which render these materials highly hygroscopic, and the presence of surface silanol groups also implies functionality to allow the better control of drug loading and release [12,13,14]. By functionalizing the surface, the new functional groups can develop strong interactions, which may lead to drug release retardation [15,16,17]. However, in an utterly different approach, the contribution of attractive drug-surface interactions is not necessarily increased, but the hydrophilicity of the silica carrier is decreased. Therefore, the penetration of the aqueous medium is decelerated, resulting in the desired controlled drug release [16,18,19]. An efficient modification route is silylation with trimethylchlorosilane (TMCS) [18] for this purpose. Several studies focus on the immediate release of poorly water-soluble drugs [20,21,22], but achieving the controlled release of highly water-soluble, hydrophilic APIs [23,24,25]—by using surface-modified MPS—as a carrier is a less studied area.

Drug loading into MPSs can be divided into two main types: solvent-free and solvent-based methods [26]. In general, solvent-based filling methods consist of two steps: the dispersion of MPS in a concentrated solution of the drug molecule and the subsequent removal of the liquid medium [13,27]. Solvent removal can be carried out by filtration (immersion method) or evaporation (spray drying, solvent drying, or incipient impregnation methods) [14]. Although the immersion method is the most widespread, a higher loading extent is more likely to be reached via solvent evaporation [20,28]. During the solvent evaporation process, the MPS suspension is stirred for a well-defined time; then the solvent is evaporated with the help of a rotary evaporator (rotavapor). As the loading solvent evaporates, a concentration gradient develops, and the incorporation of the drug into the pores is initiated.

During the incorporation process, the adsorbed API becomes amorphous on the pore surface. The inhibition of drug crystallization can be explained by two mechanisms [29]: (1) as secondary interactions develop between the drug and the functional groups of the silica surface, the Gibbs free energy of the amorphous adsorbed material is lower than that of the crystalline one; (2) the critical crystalline nucleus of the drug is smaller than the mesoporous diameter; thereby, nucleus growth is inhibited.

Among MPSs, only a limited number of studies can be found on hydrophobized Syloid-based drug delivery systems [30,31], even though these materials have highly developed mesoporous networks and a large surface area. Moreover, these systems are ideal to adsorb a remarkable amount of API. Based on the literature, they have been used mainly to enhance the dissolution of poorly water-soluble drugs. Besides, their application is safe; the No Observed Adverse Effect level (NOAEL) of orally administered Syloid 244 was high, with 9000 mg/kg body weight/day in rats with exposure of 90 days [32].

As the conventional theoretical methods struggle to deal with drug release kinetics, the development of a universal approach would be vital to this field. A promising but not yet popular approach utilizes a so-called lumped second-order kinetic model to address mixed release kinetics [33]. However, to the best of our knowledge, it has not been tested how efficiently this equation can describe the release kinetics of different, widely used release models. Its application to the release from MPSs has not been investigated either.

The systematic release control of LDME can be important in the treatment of severe PD. Therefore, we aimed to reach a possible release control route by applying the Syloid XDP 3050 drug carrier (SYL-0). Our aim was to decrease the wettability of SYL-0 reproducibly with the help of the TMCS silylating agent. Accordingly, we intended to test the feasibility of SYL-0 and its silylated derivatives for loading a well-soluble API with a narrow therapeutic window and high solubility into their mesopores in order to achieve controlled release. Levodopa methyl ester hydrochloride (melevodopa hydrochloride, LDME) was used in this work.

## 2. Materials and Methods

### 2.1. Materials

Syloid XDP 3050 (SYL-0) was manufactured and kindly provided by Grace Materials Technologies (Grace GmbH, Worms, Germany). TMCS, *n*-hexane, hydrochloride acid, monosodium phosphate, 85% *w*/*w* phosphoric acid and LDME (L-3,4-dihydroxyphenylalanine methyl ester hydrochloride), LD (L-3,4-dihydroxyphenylalanine) and 96% *w*/*w* acetic acid were manufactured by Merck Ltd. (Darmstadt, Germany). Potassium hydroxide and methanol were purchased from manufacturer VWR International Ltd. (Radnor, PA, USA).

### 2.2. 3^2^ Full-Factorial Design

After an in-depth review of the literature, the main factors influencing the pharmacokinetic properties of the ideal product were determined and summarized on a fishbone/Ishikawa diagram (Figure 1). Our goal was to prepare a stable formulation, potentially capable of treating LRMCs. According to the preliminary risk assessment, the LDME:SYL ratio and the wettability of the used MPS seemed to be the factors with the greatest influence on the drug release rate.

The main influencing factors were systematically varied as it can be seen in Table 1, thus constructing a 3^2^ full-factorial experimental design [34].

### 2.3. Hydrophobization of SYL-0

Experimental results reported in the literature showed that water plays a key role in the reaction [35]; the adsorbed water content of the silica surface has to be standardized before the reaction. The adsorbed water content of SYL-0 was standardized in a climate chamber (KKS TOP+, Wodzisław Śląski, Poland) at 40 °C and 75% relative humidity (RH) for at least 3 days to provide an equilibrated amount of adsorbed water. The hydrophobization reaction was carried out at 25 °C. A total of 0.200 g of SYL-0 was dispersed in 20.0 mL of *n*-hexane in a pre-hydrophobized capped glass vial with a volume of 40 mL. TMCS was added to the dispersion in a concentration of 10 mM (to provide the SYL-1 derivative with moderate wettability) or 20 mM (to provide the SYL-2 derivative with poor wettability). The reaction mixtures were stirred by magnetic stirring bars at 700 rpm for 2 h; then the solids content was obtained through filtration on Whatman membranes with a pore size of 0.45 µm (GE Healthcare Sciences, Chalfont St Giles, United Kingdom) and the concomitant washing with 3 × 15 mL of *n*-hexane. The reaction mechanism is summarized in Figure 2.

At the end of the process, the samples were kept at 70 °C for 3 days.

### 2.4. Characterization of Hydrophilic and Hydrophobized Silicas (SYLs)

The hydrophobicity of the carriers was evaluated; moreover, the change in the functional group composition of the surface was also proven with the help of various measurement techniques.

#### 2.4.1. Contact angle Measurement

The apparent contact angles (Θ) were measured with an OCA 20 Optical Contact Angle Measuring System (Dataphysics, Filderstadt, Germany). The carriers were compressed under a pressure of 3 tons by a Specac hydraulic press (Specac Inc., Fort Washington, MD, USA). The contact angle of the pastilles was determined by using bi-distilled water test liquid (interfacial tension of polar component (γ_p_) = 50.2 mN/m, interfacial tension of disperse component (γ_d_) = 22.6 mN/m). The measurements were carried out at 25 °C. The volume of the water droplets was 4.3 µL.

#### 2.4.2. Streaming Potential Measurement-Assisted Particle Charge Titrations

The streaming potential values in the 1 g/100 mL (1 mixed%) dispersions of the initial and the hydrophobized silica particles were measured with a PCD-02 Particle Charge Detector (Mütek Analytic GmbH, Herrsching am Ammersee, Germany). The negative surface charge excess of the silica particles was neutralized by 1 g/100 mL (1 mixed%) hexadecyl(trimethyl)ammonium bromide (CTAB) solution during particle charge titrations. The pH of the silica dispersions was set to 11.3 with the help of the NH_3_/NH_4_Cl buffer solution to ensure the complete deprotonation of the silanol groups (highly above the point of zero charge of silicas) [36].

The charge-neutral states were achieved at streaming potentials of 0 mV. The specific surface charge excess values, and therefore the specific silanol group densities, were calculated according to the following equation (Equation (1)), assuming that the positive charge of a CTAB molecule neutralizes the negative charge of a deprotonated silanol group and the adsorption of NH_4_^+^ from the buffer on the silica surface is negligible:(1)$$d_{Si-OH}=\frac{V_{CTAB}\times c_{CTAB\;}\times N_{A}}{c_{SYL}\times V_{SYL}\times M_{CTAB}\times A_{s}\times 10^{18}},$$
where *d*_Si-OH_ is the surface density of the silanol group (1/nm^2^); *V_CTAB_* and *c_CTAB_* are the volume (mL) and the concentration (g/100 mL) of the charge-neutralizing agent solution, respectively; *N_A_* is the Avogadro constant (6.022 × 10^23^ 1/mol); V_SYL_ and c_SYL_ are the volume (mL) and the concentration (g/100 mL) of the silica dispersion, respectively; MCTAB is the molar mass (364.45 g/mol) of the titrant CTAB, while A_s_ (m^2^/g) is the specific surface of the silica. The initially negative surface charge was also reversed for each sample by adding an excess amount of titrant to the initial dispersion.

### 2.5. Fourier-Transform Infrared (FT-IR) Spectroscopy

The mid-IR spectra of the SYLs (homogenized in 0.15 g KBr disks) were recorded in the spectral range of 400–4000 cm^−1^ with the help of an AVATAR 330 FT-IR spectrometer (Thermo Nicolet, Unicam Hungary Ltd., Budapest, Hungary), equipped with a deuterated triglycine sulfate detector. The spectral resolution was 2 cm^−1^, and 128 scans were performed. OriginPro 8.6 software (OriginLab Corporation, Northampton, MA, USA) was used for the spectral analysis. During sample preparation, the pressure was 10 tons, and the diameter of the pressings was 13 mm; a Specac Hydraulic Press was used (Specac Inc., Orpington, UK). Hydrophobization with TMCS and the effect of loading on the secondary interactions between LDME and the silica were evaluated by FT-IR as well.

### 2.6. Loading of LDME into SYL via Rotavapor

For the loading process, Rotavapor R-125 equipment (BÜCHI Labortechnik AG, Flawil, Switzerland) was used. A different amount of LDME (m_LDME_ = 25, 62.5 or 100 mg) was dissolved in 5.00 mL of methanol in a round-bottom flask. Thereafter, SYL-0 or SYL-1 or SYL-2 was dispersed in the solution according to the planned SYL:LDME mass ratio in the products (see in Table 1). The sum of the mass of the dissolved LDME and dispersed SYL was around 500 mg in every case. Methanol was evaporated at 50 °C and 135 mbar pressure. The applied rotation speed was 120 rpm. Due to the rotation, the dispersion remained homogenous during the process. The resulting dry powder was stored in a freezer.

The physical stability of the powders was investigated under different conditions, stored at −20 °C and at 40 °C and 75% RH for 3 months, based on the ICH Q1A (R2) guideline proposal [37].

LDME and the given SYL were mixed and homogenized in a Turbula mixer (Turbula System Schatz; Willy A. Bachofen AG Maschinenfabrik, Basel, Switzerland) for 10 min to make physical mixtures. These powders were used as a reference in the measurements.

### 2.7. BET Measurements

Nitrogen adsorption isotherms were recorded at 77 K using a QuantaChrome Nova 3000 surface area analyzer. Before the measurement, the samples were outgassed at 403 K for 1 h to remove any adsorbed contaminants. The specific surface areas (as BET) were calculated using the multipoint BET method on the basis of six data points of the adsorption isotherms near monolayer coverage.

### 2.8. Residual Solvent Content of the Products by Gas Chromatography (GC)

The methanol content of the samples was analyzed by a gas chromatograph (Shimadzu GC-14B) equipped with a thermal conductivity (TCD) and a flame ionization detector (FID). The calibration curve was previously determined in the range of 0–0.63 g/mL of methanol in ethanol medium. The concentration of methanol is directly proportional to the area of the peak for methanol according to the y = 9,910,733 × x equation (R^2^ = 0.9897). The unit of y was µV, and the unit of x was g/mL. The retention time of methanol was 1.49 min; the retention time of ethanol was 3.25 min. The ICH Q3C (R8) guideline sets the limit for methanol at 3000 ppm (*w*/*w*) [38].

### 2.9. Raman Spectroscopy

The distribution of LDME in SYLs was investigated by Raman chemical mapping using a Thermo Fisher DXR Dispersive Raman instrument (Thermo Fisher Scientific Inc., Waltham, MA, USA), equipped with a CCD camera and a diode laser, operating at a wavelength of 780 nm. Raman measurements were carried out with a laser power of 24 mW at a slit aperture size of 50 µm. A surface with a size of 100 µm × 100 µm was analyzed with a step size of 10 µm, with an exposure time of 2 s and an acquisition time of 2 s, for a total of 6 scans per spectrum in the spectral range of 3500–200 cm^−1^ with cosmic-ray and fluorescence corrections. The Raman spectra were normalized to eliminate the intensity deviation between the measured areas.

### 2.10. X-ray Powder Diffractometry (XRPD)

The X-ray diffractograms were recorded with a Bruker D8 Advance diffractometer (Bruker AXS GmbH, Kalsruhe, Germany) with Cu K lI (λ = 1.5406 Å) and a VANTEC detector. The samples were scanned at 40 kV voltage and 40 mA current. The angular range was between 5 and 40° (2θ); the interval was 0.007°; the step time was 0.1 s. The samples were placed in a quartz holder at ambient temperature and RH. The K_α2_ radiation was stripped from the diffractogram and smoothening and evaluation were performed with DIFFRACTPLUS EVA 5.2.0.3 software (Bruker AXS, Kalsruhe, Germany). Overall, the success of loading, the physical stability of the reference and the samples could be followed by XRPD.

### 2.11. Differential Scanning Calorimetry (DSC) Measurements

Then, 3–5 mg samples were measured with Mettler-Toledo DSC-821e equipment (Greifensee, Switzerland). The samples were placed in a sample holder made from aluminium. The products were investigated in the temperature range of 25–350 °C with a heating rate of 10 °C/min. Each sample was normalized to the sample size. The DSC curves were analyzed with the help of STARe 9.30 software (Mettler-Toledo GmbH, Gießen, Germany).

### 2.12. Particle size Analysis

The LEICA Image Processing and Analysis System (LEICA Q500MC, Leica Microsystems Cambridge Ltd., Cambridge, United Kingdom) was used to measure the particle size of the products and the SYL carriers. Particle size was determined based on the length value in the output file.

### 2.13. High-Performance Chromatography (HPLC) Method

The LDME and accidental LD content of the products were analyzed with a 1260 HPLC Agilent system (Agilent Technologies, San Diego, CA, USA). The mobile phase consisted of acetate buffer (pH = 5.0):methanol = 90:10 (*v*/*v*). A total of 1.0 L of acetate buffer was composed of 0.714 cm^3^ of 96% *m/m* acetic acid solution and 0.561 g of potassium hydroxide and MilliQ water. A Chrome-Clone 5 µm C18 100 Å column (150 × 4.6 mm, Phenomenex, Torrance, CA, USA) was applied, connected with a C-18 security guard cartridge. A total of 10 µL of the sample was injected, and isocratic elution was applied with 1 mL/min flow for 8 min at 30 °C. Data were evaluated with ChemStation B.04.03. software (Agilent Technologies, Santa Clara, CA, USA). Both APIs were analyzed at 280 nm with a diode array detector.

Using the optimized chromatographic conditions of separation, the validation properties of LD and LDME were defined (Table 2).

### 2.14. Determination of API Content

The LDME content was determined after extraction. During the extraction process, 10 mg of the products were dispersed in 10 mL of methanol in a centrifuge tube; the dispersions were left at room temperature for 1 h; then they were centrifuged in a HERMLE Z323 K high-performance refrigerated centrifuge (Hermle AG, Gosheim, Germany) at 13,500 rpm for 10 min. Altogether, 28 mL of supernatant were collected. The procedure was repeated 3 times. The residual liquid was filtered using a syringe. The supernatants of the 3 cycles were mixed, and the API content was determined by HPLC.

### 2.15. Release Studies

Phosphate buffer was used as a dissolution medium: 1.0 L of the buffer solution contained around 1.7 g of NaOH and 6.8 g of KH_2_PO_4_. The pH was set to 6.8 ± 0.1 with 85% *w*/*w* phosphoric acid or 1 M NaOH solution when needed.

The dissolution tests and the degradation studies of the API were performed in a Hanson SR8 Plus release device (Hanson Research, Chatsworth, CA, USA) in 100 mL of pH = 6.8 buffer at 37 °C. The stirring rate of the simple paddle was 100 rpm. A total of 3 mL of samples was taken at predetermined time intervals (at 5, 10, 15, 30, 60, 90 120, 180, 240, 300 min), and the volume was replaced with 3 mL of fresh dissolution medium. A total of 90 mg of the binary system (containing SYL and LDME powder) was filled into a hard gelatin capsule (capsula operculata 00, Molar Chemicals, Budapest, Hungary). The capsule was placed in a seamless cellulose dialysis tubing (length: 8 cm, average flat width: 23 mm, Merck Ltd., Budapest, Hungary).

As LDME is chemically unstable in aqueous media and is converted into LD [39] during the release studies, it was necessary to follow the release kinetics with HPLC to quantify both LDME and LD; the released amount of LDME was defined according to Equation (2):(2)$$n_{LDME,\;released}=n_{LDME,\;measured}+n_{\;LD,\;measured},$$
where *n_LDME, released_* means the released moles of LDME; *n_LDME, measured_* and *n_LD, measured_* are the measured moles of LDME and LD, respectively.

The release kinetics of LDME from the products were determined. The fitting of the results was tested with zero order, first order, Higuchi, the Korsmeyer–Peppas model and the Hixson–Crowell cube root law [40]. The model for the fitting with the largest R^2^ was accepted as the release kinetics.

A dissolution can be described by a saturation curve; thus, Equation (3) can be used [33]:(3)$$\frac{m_{LDME,\;released}}{m_{LDME,\;loaded}}=\frac{a*t}{1+b*t},$$
where time is t (min) and we look for *parameters a* (1/min) and *b* (1/min) with which the equation best fits the measured points. *Parameters a* and *b* were determined by iteration using the least-squares method (Solver extension of Excel 2016, Redmond, WA, USA). This equation was proposed mainly for “lumped” second-order kinetics but we would like to investigate its utility for different release kinetics. It is advantageous to use this equation because it is model-independent and when the release is not complete, the application of this model does not distort the results.

If the limit of the equation is t = 0 (Equation (4)):(4)$$\underset{t\to 0}{lim}\frac{a\times t}{1+b\times t}=a\times t=\frac{m_{LDME,\;initially\;released}}{m_{LDME,\;loaded}}$$

Thus, *parameter a* corresponds to the initial release rate (IRR).

Thereafter, the similarity factor (f_2_) [41,42] between the fitted model and the actual release results was calculated.

f_2_ is the logarithmic reciprocal square root transformation of the sum of squared error and is a measurement of the similarity in the percent (%) dissolution between the two curves, i.e., it can be defined in the following way (Equation (5)):(5)$$f_{2}=50\left[\mathrm{lg}1-\frac{1}{n}\overset{n}{\underset{t=1}{\sum }}{\left(R_{t}-T_{t}\right)}^{2}{)}^{-0.5}\times 100\right]$$
where *n* is the number of observations; *R_t_* is the mean percentage of drug dissolved from the reference formulation (in our case: the percentage of the model curve); *T_t_* is the percentage of drug dissolved from the test formulation. As the value of f_2_ approaches 100, the two compared curves are getting similar.

### 2.16. Investigation of Degradation Kinetics of LDME

As it is known from the literature that LDME has a tendency to degrade in aqueous media, therefore its stability and degradation kinetics were also determined. A total of 40 ± 1 mg of API was dissolved in the dissolution medium (pH = 6.8 phosphate buffer) which was thermostated at 37 °C. Samples were taken at 0, 15, 30, 60, 90, 120, 180, 240, 300 min quantified by using HPLC with the method detailed in Section 2.13. Three parallels were measured.

### 2.17. Statistical Analysis of the Results

The contact angle of the SYLs was followed with a focus on the mean and standard deviation. The effect of standardizing the reaction on standard deviation was determined by a 2-sample variance test. The change in particle size after loading a SYL with API was also investigated by a 2-sample *t*-test. All these tests were conducted by Minitab 17 Statistical Software (Minitab Ltd., Coventry, UK). Besides, the statistical comparison of the release results of unique products was carried out by using the Tukey HSD test (*n* = 3). A certain phenomenon was considered significant when *p* < 0.05.

To investigate the quadratic response surface and to construct a polynomial model based on the *parameter a* values of the release results, TIBCO Statistica 13.4 (Statsoft Hungary, Budapest, Hungary) statistical software was used. The significance of the variables and interactions based on their holistic effect was evaluated using analysis of variance (ANOVA). Differences were considered significant when *p* < 0.05.

## 3. Results and Discussion

### 3.1. Evaluation of the Hydrophobization Reaction

To quantify the efficacy of carrier hydrophobization, we examined the surface of the hydrophobized silica particles applying contact angle measurements, particle charge titrations and infrared spectroscopy.

At first, the optimization of the hydrophobization reaction was carried out. The adsorbed water amount of SYL-0 proved to be a key factor because it might affect the hydrolysis of the functionalizing agent; therefore, the storage circumstances were standardized (SYL-0 was kept under 40 °C and 75% RH for 3 days). Besides, the containers have to be hydrophobized prior to the reaction to ensure reproducibility via preventing the unwanted reactant consumption by the -OH content of the glass. As a result of these preliminary measures, the standard deviations of the contact angles of SYL-1 and SYL-2 showed a significant decrease (*p* < 0.05). The evaporation of water content at 110 °C resulted in the formation of the so-called SYL-DRY product. The contact angle of SYL-DRY was 0°, as well as in the case of raw SYL-0 (kept under 40 °C and 75% RH for 3 days), which also confirmed the role of water in the reaction.

During surface functionalization, the water contact angles increased with increasing TMCS concentration (Table 3), which can be translated to a decrease in wettability.

As the contact angles of SYLs were reduced, the water droplets had a lower tendency to spread on the pastille surface. The time-dependency of the water contact angles can be seen in Figure 3.

The CTAB concentrations belonging to 0 streaming potential could give an estimation for the surface density of silanol groups. The concentration of TMCS in the reaction mixture reduced d_Si-OH_, indicating that the Si-OH groups were functionalized during the reaction (Table 4).

On the other hand, it was necessary to prove that -Si(CH_3_)_3_ groups were covalently attached to the surface as a result of the reaction, and the surface density of silanol groups did not decrease because of the storage temperature of 110 °C after the reaction. The functionalization of SYL was proved by IR spectroscopy. The absorption of the samples in the mid-IR range could provide qualitative information on the changes in functional group composition. Figure 4 shows a comparison of the spectra of SYL-1, SYL-2 and SYL-0. According to the analysis, the spectra of both functionalized silicas (SYL-1 and SYL-2) showed the same types of changes.

The band at 474 cm^−1^ could be assigned to Si-O-Si bending vibrations, while at 804 cm^−1^ it indicates the symmetric stretching vibrations of Si-O-Si. The very strong and broad IR band located at 1105 cm^−1^ with a shoulder-type peak at 1188 cm^−1^ can be assigned to the transversal optic and longitudinal optic modes of the Si-O-Si asymmetric stretching vibrations [43]. As the reaction left the siloxane groups intact, the relative intensity of the bands did not differ. The band at 972 cm^−1^ corresponded to the stretching vibrations of the Si–OH groups [44]. The intensity of this band decreased drastically in the case of SYL-1 and SYL-2, compared to SYL-0, indicating a significant decline in the density of surface silanol groups. The presence of the formed Si-CH_3_ bonds on the surface was confirmed by the new peaks in the range of 865–750 cm^−1^ (peaks: 759 cm^−1^, 850 cm^−1^ and 865 cm^−1^) [45]. Meanwhile, the band at 1633 cm^−1^ belongs to the bending vibration, and the peak at 3435 cm^−1^ belongs to the stretching vibration of adsorbed H_2_O molecules [43]. The intensity of both bands decreased in the SYL derivative spectra despite the fact that both samples were stored at 70 °C, at the same RH, which means that the hydrophobized SYL could adsorb a smaller amount of water. This can be attributed to the lower surface -OH content, which otherwise could render the surfaces able to make strong H-bonds. Besides, new peaks appear at 2906 cm^−1^ and 2964 cm^−1^ which are corresponding to the C-H vibration of the trimethylsilyl groups.

Based on the diffractograms, it was proven that amorphous mesoporous silica particles were used as raw material, and during the hydrophobization reaction, this amorphous nature remained unchanged (see SYL-0, SYL-1 and SYL-2 in Figure 5C). The storage at 70 °C did not affect the surface area, which is in good accordance with the literature (sintering occurs only above 550 °C) [46].

### 3.2. Loading of LDME into the Pores of SYLs

A total of 5–20% *w*/*w* of LDME was loaded into the SYL pores. The characteristic peaks of the physical mixtures (physical mixtures were prepared in the 5–20% *w*/*w* range) and the raw LDME on the diffractogram are shown in Figure 5A. As a result of the loading process, the API became amorphous, which was indicated by the absence of the characteristic crystalline peaks of the API on the diffractograms (Figure 5B). DSC was also used as an additional confirmatory measurement technique to check the success of LDME loading into the mesopores of SYLs. As a result of amorphization, the endothermic peak related to the melting of the crystalline API (Figure 5C) disappeared in the case of the products (Figure 5D) in the investigated range. The methanol content of all products was lower than the requirement of the ICH Q3C (R8) guideline (Appendix A, Table A1).

### 3.3. Interaction between the Silica Surface and LDME

The LDME-containing products were also analyzed with FT-IR to determine the accidental secondary interactions between the drug and SYL. In the products, even in 20% *w*/*w* API-containing products, most bands of the active substance were shaded by the signal of SYL. In all products, ν_C=O_ of LDME shifted from 1736 cm^−1^ to a higher wavenumber, which may indicate developing a secondary interaction between the surface and the API; however, it might be weaker than it was between the LDME molecules. ν_Si-OH_ shifted from 964 cm^−1^ to a lower wavenumber, presumably due to new intermolecular interactions in the case of SYL-0-containing products, but this phenomenon was not observed in the hydrophobized SYL-containing samples, probably due to the lower amount of silanol groups; other bands covered the signal, and fewer secondary interactions were present.

### 3.4. Distribution of LDME in the Products, Its Effect on Particle Size and BET Specific Surface Area

Raman mapping was carried out to determine the distribution of LDME in different SYL products (Figure 6). For this purpose, the Raman spectrum of pure LDME was used as a reference to visualize the chemical map of the products showing the amount of LDME. Not surprisingly, increasing the amount of LDME during the loading process resulted in a higher final drug content, as evidenced by the greater extent of regions on the chemical map, marked by red colour. The effect of hydrophobization was remarkable only in the case of low drug content: the Raman map of SYL 1-5 and SYL 2-5 indicates that LDME can be found only sporadically in comparison to SYL 0-5, which can be attributed to the higher extent of interactions between the polar API and the polar functional groups of the surface. In the hydrophobic products, the API might develop a stronger interaction between the API molecules than between the surface APIs.

The study of the products and the SYL carriers proved that LDME was likely to be loaded into the pores and did not recrystallize on the SYL surface, as no significant change was detected among the samples compared to the given SYL (Table 5). Every product was compared to the SYL, which had the same hydrophobicity as the silica used in the product (for example, SYL-2-12.5 was not significantly different from SYL-2). The silylation process resulted in a decreased nitrogen adsorption capacity, i.e., decreased specific areas (Table 5). Thus, the BET-specific surface area (A_s_) of the original SYL-0—283 m^2^/g—decreased to approximately 244 m^2^/g in SYL-1, 235 m^2^/g in SYL-2. This phenomenon had been observed in the literature earlier after the hydrophobization of mesoporous silica with TMCS [47]. Besides, the BET surface area also decreased after the incorporation of the API into the pores. These results also confirm the success of the loading process.

### 3.5. Stability of the Products

The chemical stability of the powders was appropriate upon storage in the freezer; however, the powders at 40 °C, 75% RH became light brown. According to the additional HPLC measurement, drug content decreased in these samples, indicating that the LDME-containing samples should be stored in the freezer just as the API itself. However, the residual API remained amorphous in the products within the 5–20% *w*/*w* range of the initial API content.

It makes sense to interpret the results of the physical stability investigation only in such cases where the API did not suffer chemical decomposition. Thus, only the results of samples stored at −20 °C are shown. A weak sign of crystalline LDME appeared in the X-ray diffractogram in the case of the SYL-0-20 product after 3 months at −20 °C, which probably means that the 20% *w*/*w* LDME ratio could be too high for all API molecules to be incorporated into the pores. Thus, the amorphous API remained stable in the SYLs at lower than 20% *w*/*w* LDME content.

In comparison, LDME was directly amorphized in the sample holder of XRPD under the same conditions as used in the rotavapor. The amorphized raw LDME recrystallized due to weak scraping and after leaving it untouched for one day under the conditions of the stability tests. The amorphization kinetics of the pure LDME was investigated at 40 °C, 75% RH. The recrystallization process took place in about 100 min under these circumstances (Figure 7).

Compared to the reference, the recrystallization rate of the products turned out to be slower by orders of magnitude. The loading of LDME into the pores of MPSs seems to be a quite reliable tool to prevent recrystallization if the% *w*/*w* of the API is under a certain ratio. Thus, it is important to determine the storage conditions and the related maximum drug loading below which the API remains in the mesopores and in an amorphous state, resulting in the desired stability of the products.

### 3.6. Degradation of LDME at pH = 6.8

As LDME is absorbed in the small intestines, a pH = 6.8 ± 0.1 phosphate buffer solution was used as the dissolution medium. The release was carried out immediately after the silica loading. As the main degradation product of LDME in aqueous medium is LD, and a significant amount of decomposition product was expected to be formed during the dissolution tests, LDME and LD were quantified, and the released amount of LDME was equal to the amount of drug dissolved. It is known from the literature that LDME suffers degradation according to first-order kinetics [39]. The degradation rate was measured under the dissolution conditions (Figure 8).

The reaction rate coefficient (k) was 2.008 × 10-3 ± 2.94 × 10^−4^ 1/min, whereas the half-time (t_1/2_) was 345.1 ± 50.5 min; therefore, the actual concentration of LDME can be described by Equation (6):(6)$$\left[LDME\right]={\left[LDME\right]}_{0}\times e^{-2.008\times 10^{-3}\frac{1}{min}\times t}$$

### 3.7. Release Studies

Dissolution kinetics were investigated for 5 h. The resulting curves are shown in Figure 9. As a general tendency, the mean release rate of LDME could be distinguished according to the hydrophobicity of SYL. As hydrophobicity increased, the release rate showed an unambiguous decrease. There was a spectacular difference detected between the SYL-2-containing samples and the other two samples; however, the difference was noticeably smaller in the case of 20% *w*/*w* LDME-containing samples. A probable explanation for this is the much poorer aqueous wetting of SYL-2 compared to the other two SYLs.

The kinetics of the release curves were determined. The first-order kinetics of the reference LDME was not changed in the case of dissolution from SYL-0-20, SYL-1-20 and SYL-2-20 products. It was probably due to the high loading content. Thanks to this, the diffusion of water molecules was not really hindered in the pores at the beginning, and they could contact the API molecules easily, as polar API molecules covered the pore surface presumably thickly. Thus, the relative increase in SYL and the decrease in LDME resulted in the change of release kinetics. The release of SYL-0-5, SYL-0-12.5, SYL-1-5 and SYL-1-12.5 products fitted to the Hixson–Crowell model the most closely. It could be due to the decrease in the surface area as a consequence of the release of the API. Compared to these, the release of SYL-2-5 and SYL-2-12.5 products was zero order; therefore, SYL-2 with API content in the range of 5-12.5 *w*/*w*% may be appropriate for maintaining a constant blood level, but if the product contains 20% *w*/*w* LDME, it will not be proper for this purpose.

To interpret the results statistically, *parameters a* and *b* were determined based on the procedure detailed in Section 2.15. In the following part, we would like to focus on *parameter a,* which characterizes IRR, i.e., as this parameter increases, so does the drug release rate. The parameters of the release curves can be seen in Table 6. After the determination of *parameters a* and *b*, which can model the release kinetics, the similarity factor (f_2_) was defined between the model and the actual release results. According to this value, the models fitted close to the actual release results; therefore, we suggest that this model be used in the case of products with such release kinetics during the evaluation of drug release. Figure A1 presents the methodology of the *parameters a* and *b* showing the model curve (based on Equation (3)) and the actual release results of SYL-0-20. Besides, the curve based on the *a* × t (Equation (4)) is also shown which fits the initial stage of the model curve.

Based on the results, a polynomial equation was constructed to give the influence of the independent variables (c_TMCS_ and LDME% *w*/*w*) on the dependent variable (IRR).

The IRR of LDME can be described by Equation (7):*Parameter a = 9.423 × 10^−3^ − **7.818** × **10^−3^ x −** 9.1 × 10^−5^ x^2^ + **1.921** × **10^−3^ y − 1.271** × **10^−3^ y^2^ −** 6.55 × 10^−4^ x y + 2.33 × 10^−4^ x y^2^ + 8.15 × 10^−3^ x^2^ y **−** 6.23 × 10^−4^ x^2^ y^2^*(7)
where *x* belongs to the TMCS concentration; *y* belongs to the LDME mass percent prior to the preparation of the product. The members of the equation highlighted in bold have a significant influence on IRR. The statistical parameters of this equation are: R^2^ = 0.93462, adjusted R^2^ = 0.90556, mean square residual = 0.0000048. By the removal of the *y^2^*, the *x y^2^* member from the polynomial equation fits the most closely to the measured points. As a consequence, there is a negative linear relationship between the IRR and the hydrophobization extent. After the removal of these points, the result is Equation (8):*Parameter a = 9.423 × 10^−3^ − **7.818** × **10^−3^ x + 1.921** × **10^−3^ y** − **1.271** × **10^−3^ y^2^** − 6.55 × 10^−4^ x y + 8.15 × 10^−3^ x^2^ y − 6.23 × 10^−4^ x^2^ y^2^*(8)

The statistical parameters of this equation are: R^2^ = 0.93381, adjusted R^2^ = 0.91395, mean square residual = 0.0000044. The members of the equation highlighted in bold have a significant influence on IRR. This equation defines the surface plot below (Figure 10).

Based on the IRR results, we can see that both factors have a significant linear effect on the release rate. Hydrophobization had a significant retarding effect on release. The results also indicate that LDME% *w*/*w* had a positive significant effect and a negative quadratic effect on *parameter a*.

According to the Tukey HSD results (see Table A2), the release rate decreased significantly in all products compared to the LDME reference except for SYL-0-20. In the case of samples with the same LDME content and different wettability, the increasing degree of hydrophobization had a significant retarding effect on LDME release. *Parameter a* decreased significantly for SYL-2-containing products at a certain API ratio compared to the SYL-0-containing products; however, there was no significant difference between the SYL-0 and SYL-1 samples, which indicates that 10 mM of TMCS was not enough to decelerate the release rate, but 20 mM was enough.

In contrast to the comparison of individual values with the help of the Tukey HSD test, according to the result of the ANOVA evaluation of the 3^2^ full-factorial design, the API% *w/w* also had a significant effect on *parameter a,* but it does not seem so obvious when the individual IRR values are compared. This finding emphasizes the importance of the methodology of the ANOVA evaluation, as it holistically compares all products and not only individual values. Summarizing the data, even if the product is stable, the ratio of the API and the hydrophobicity of the mesoporous carrier must be taken into consideration when the drug release rate is supposed to be regulated in mesoporous silica systems because it has a significant effect on the release properties.

## 4. Conclusions

In this work, the drug release control from loaded hydrophilic and hydrophobized MPSs was investigated. MPS was hydrophobized by TMCS; the model API was LDME. To analyze the release properties of hydrophobized and non-hydrophobized MPS, a 3^2^ full-factorial design was set and executed. The independent variables were the TMCS concentration (0, 10, 20 mM) and the LDME mass percent at the sample preparation (5, 12.5 and 20% *w*/*w*). The wettability of MPSs was reduced by using TMCS; the silanol group density decreased, and the trimethylsilyl groups were attached to the surface. LDME was loaded into the mesopores by a solvent drying method using a rotavapor. The API was homogenously distributed; secondary interactions were developed between the surface and the drug. The surface-drug interaction seemed to be stronger when the SYL was more hydrophilic, presumably causing sporadic LDME in the case of SYL-1-5 and SYL-2-5 compared to SYL-0-5. The drug release from the powder could be significantly decreased by the hydrophobization of MPS, verified by the evaluation of ANOVA and the Tukey HSD tests. The release rate could be significantly decreased in every product compared to the LDME reference powder, except for SYL-0-20. The release rate was evaluated by fitting to kinetic models and *parameter a,* which characterized the IRR. The adaptability of this model was also verified for different release kinetics with the help of the similarity factor which was 99.91 or higher, showing that it can be used in the case of different release kinetics, but the similarity factor is suggested to check between the actual release values and the points of model curve. The LDME release significantly decreased due to hydrophobization (*p* < 0.05) in the 5–20% *w*/*w* LDME range. However, the physical stability of the products was not sufficient when the API mass percent was 20% *w*/*w*. In the case of the SYL-0-20 product, a weak sign of crystalline fraction appeared in the X-ray diffractogram after 3 months of storage at −20 °C, while the API was not chemically stable at 40 °C, 75% RH. Higher amounts of LDME increased the value of IRR, i.e., drug release; however, the physical stability of the material loaded into the mesopores limits scientists and developers in preparing MPS-containing products with a high API content.

The use of hydrophobized MPSs can be recommended for APIs with a narrow therapeutic index because the API release can be systematically regulated by controlling the wettability of MPSs and the API content. The optimized formulations with zero order release kinetics can be ideal for preparing a formulation that can provide steady blood levels resulting in the shortening of off periods, i.e., reducing the side effects compared to traditional levodopa treatment. The same approach is used in the case of LD/carbidopa-containing duodenal infusion [48].

## Figures and Tables

**Figure 1 pharmaceutics-13-01039-f001:**
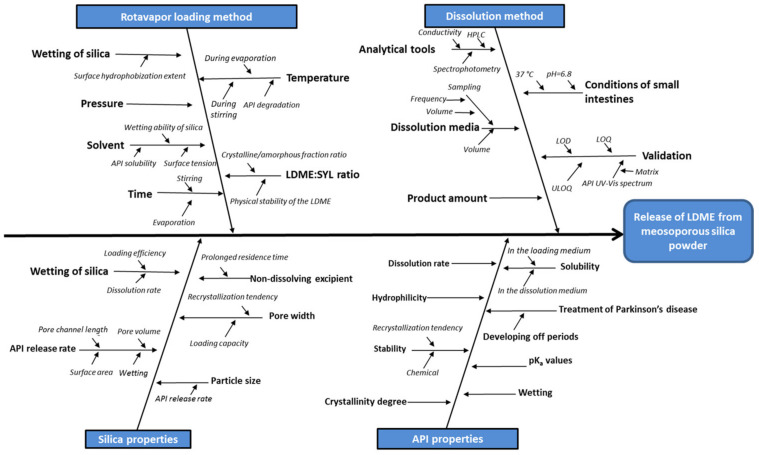
Factors influencing the properties of the product.

**Figure 2 pharmaceutics-13-01039-f002:**
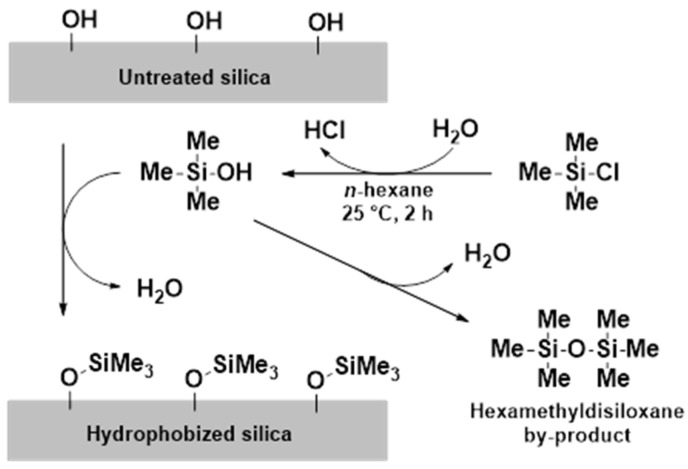
Mechanism of silica hydrophobization via trimethylchlorosilane (TMCS).

**Figure 3 pharmaceutics-13-01039-f003:**
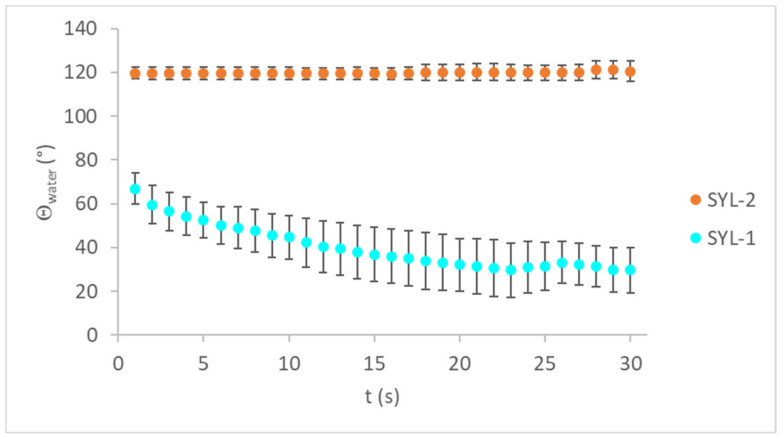
Time-dependency of water contact angles on SYL-1 and SYL-2 pastilles.

**Figure 4 pharmaceutics-13-01039-f004:**
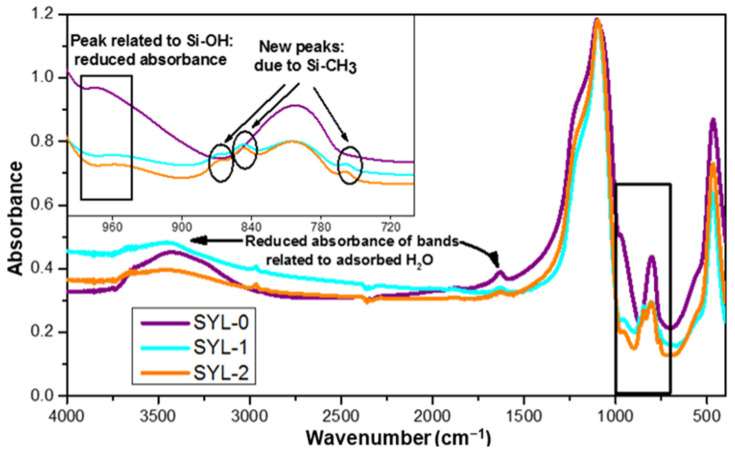
Mid-IR spectra of SYL-0, SYL-1 and SYL-2 with the 700–1000 cm^−1^ region magnified.

**Figure 5 pharmaceutics-13-01039-f005:**
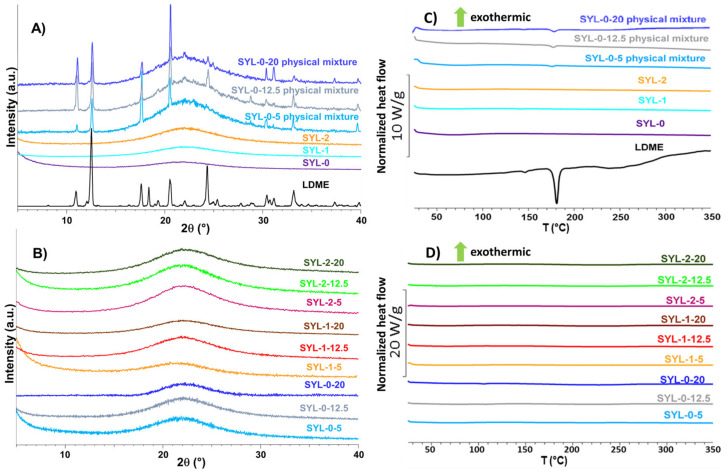
The diffractograms (**A**) and the DSC curves (**C**) of the reference materials compared to the diffractograms (**B**) and the DSC curves (**D**) of the products.

**Figure 6 pharmaceutics-13-01039-f006:**
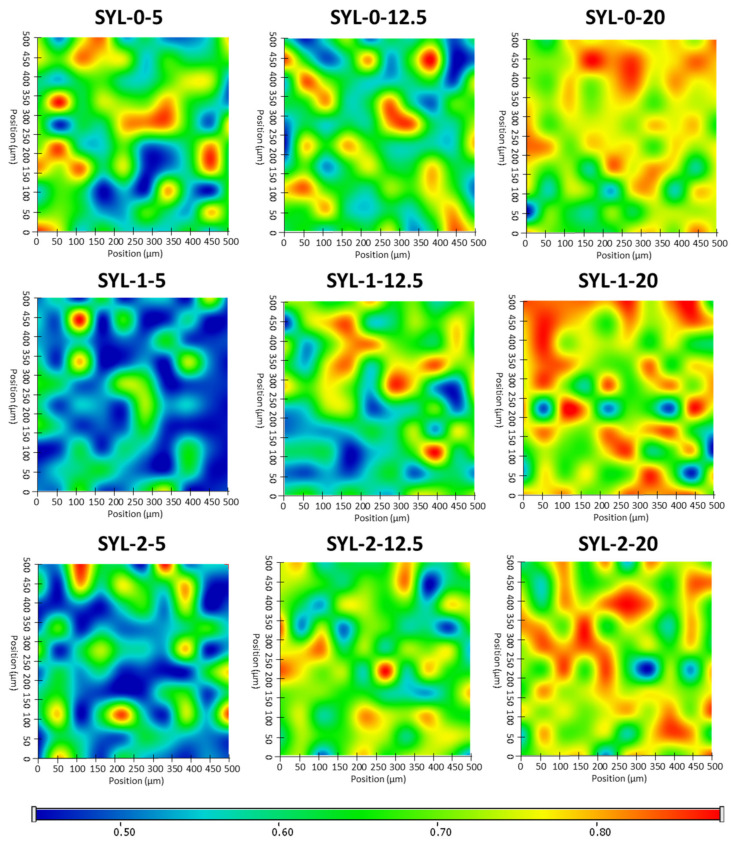
Raman chemical maps of SYLs indicating the distribution of LDME in the products.

**Figure 7 pharmaceutics-13-01039-f007:**
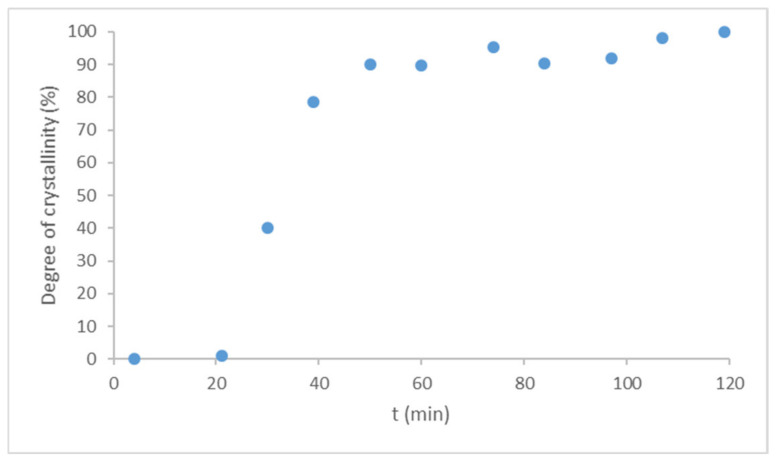
Recrystallization kinetics of pure LDME from the amorphous form at 40 °C, 75% RH under the storage conditions of the accelerated stability tests (based on the ICH Q1A (R2) guideline).

**Figure 8 pharmaceutics-13-01039-f008:**
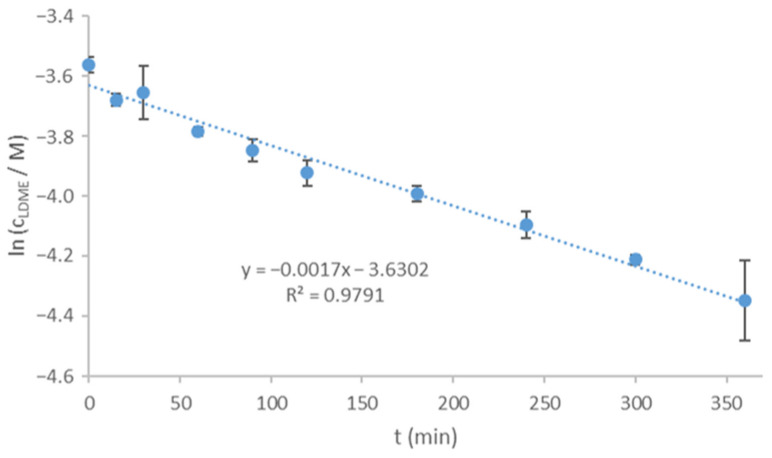
Degradation kinetics of LDME (pH = 6.8 phosphate buffer, T = 37 °C, 3 parallels).

**Figure 9 pharmaceutics-13-01039-f009:**
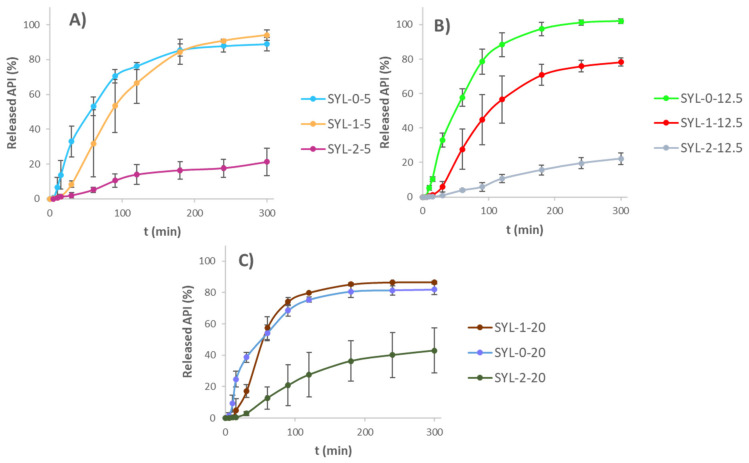
Drug release from the SYL-LDME products (**A**) 5% *w*/*w* LDME-containing products, (**B**) 12.5% *w*/*w* LDME-containing products, (**C**) 20% *w*/*w* LDME-containing products.

**Figure 10 pharmaceutics-13-01039-f010:**
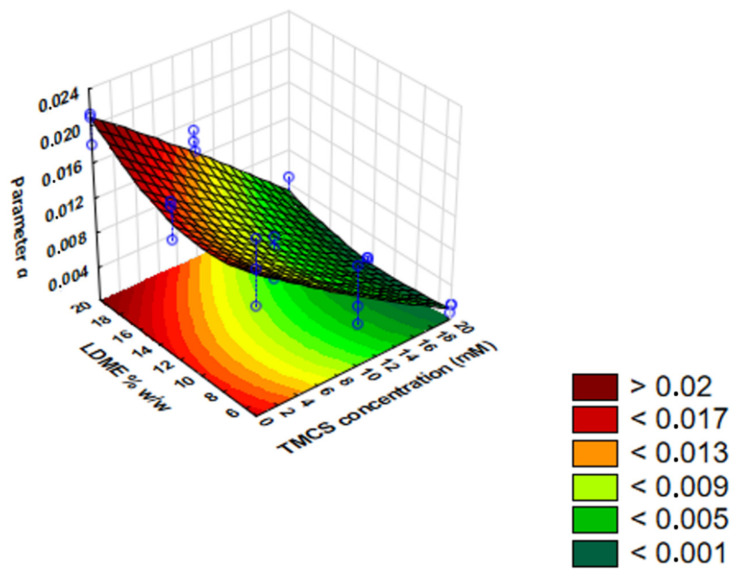
The surface plot of initial release rate (IRR) as a function of LDME% *w*/*w* and TMCS concentration.

**Table 1 pharmaceutics-13-01039-t001:** The 3^2^ full-factorial experimental design comprising the 2 main factors and their levels: −1, 0 and +1.

Amount of Hydrophobization Agent		Ratio of the API at Solvent Evaporation		Nomenclature of the Product
Level	c_TMCS_ (mM)	Level	LDME Mass Percent (% *w*/*w*)	General Format: SYL-X-Y
−1	0	−1	5	SYL-0-5
		0	12.5	SYL-0-12.5
		+1	20	SYL-0-20
0	10	−1	5	SYL-1-5
		0	12.5	SYL-1-12.5
		+1	20	SYL-1-20
+1	20	−1	5	SYL-2-5
		0	12.5	SYL-2-12.5
		+1	20	SYL-2-20

**Table 2 pharmaceutics-13-01039-t002:** Validation parameters of levodopa (LD) and levodopa methyl ester hydrochloride (LDME) using isocratic separation with HPLC-DAD system. (In the calibration equation, the unit of y was mAu × min; the unit of x was mg/mL).

API/ Validation Parameter	LD	LDME
Calibration equation	*y = 7566 × x − 3.51*	*y = 5689 × x − 5.45*
LOD (ng/mL)	8.10	37.2
LOQ (ng/mL)	23.5	112
R^2^	0.99995	0.99991
t_ret_ (min)	2.18 ± 0.003	6.610 ± 0.089

**Table 3 pharmaceutics-13-01039-t003:** Characteristic contact angles of initial and hydrophobized Syloid XDP 3050 (SYL-0) and its hydrophobized derivatives (SYL-DRY, SYL-1 and SYL-2).

Silica Types	Θ_H2O_ (°)
SYL-0	0
SYL-DRY	0
SYL-1	66.86 ± 7.16
SYL-2	120.49 ± 2.78

**Table 4 pharmaceutics-13-01039-t004:** Surface density of silanol groups on SYL surfaces.

SYL Type	SYL-0	SYL-1	SYL-2
d_Si-OH_ (1/nm^2^)	0.64	0.54	0.15

**Table 5 pharmaceutics-13-01039-t005:** Mean particle size, BET specific surface area of the products and the SYL carriers and mass ratio of LDME in the products.

Product	Mean Particle Size (µm)	A_s_ (m^2^/g)	w_LDME_ (g/g)
SYL-0	51.1 ± 15.8	283	-
SYL-0-5	46.5 ± 12.0	263	0.0479
SYL-0-12.5	55.9 ± 14.4	255	0.124
SYL-0-20	51.1 ± 19.2	208	0.214
SYL-1	49.3 ± 19.2	244	-
SYL-1-5	53.7 ± 11.4	240	0.0474
SYL-1-12.5	55.2 ± 11.1	184	0.129
SYL-1-20	51.3 ± 17.5	181	0.205
SYL-2	50.8 ± 12.4	235	-
SYL-2-5	47.3 ± 13.9	226	0.0495
SYL-2-12.5	50.9 ± 15.6	173	0.126
SYL-2-20	54.2 ± 12.8	169	0.210

**Table 6 pharmaceutics-13-01039-t006:** Parameters determining LDME release from the products and the reference, *parameters a, b*, the strongest correlated kinetic model, the R^2^ of this kinetic model with the strongest correlation and the similarity factor between the model release curve and the actual release values.

Sample	*Parameter a* (1/min)	*Parameter b* (1/min)	Release Kinetics with the Strongest Correlation	R^2^ of the Kinetic Model with the Strongest Correlation	*f_2_*
LDME (reference)	0.02659 ± 0.00327	0.02165 ± 0.00299	First order	0.9866	99.99
SYL-0-5	0.01600 ± 0.00356	0.01363 ± 0.00347	Hixson–Crowell	0.9927	99.98
SYL-0-12.5	0.01585 ± 0.00228	0.01116 ± 0.00180	Hixson–Crowell	0.9848	99.96
SYL-0-20	0.02006 ± 0.00185	0.01993 ± 0.00305	First order	0.9815	99.98
SYL-1-5	0.007694 ± 0.003309	0.004263 ± 0.00337	Hixson–Crowell	0.9781	99.95
SYL-1-12.5	0.0065001 ± 0.002508	0.004281 ± 0.002743	Hixson–Crowell	0.9781	99.96
SYL-1-20	0.01371 ± 0.00122	0.01103 ± 0.00144	First order	0.9496	99.91
SYL-2-5	0.001359 ± 0.000524	0.002931 ± 0.000712	Zero order	0.9727	100.0
SYL-2-12.5	0.0008355 ± 0.0001748	0.002598 ± 0.000229	Zero order	0.9806	100.0
SYL-2-20	0.002804 ± 0.001652	0.002509 ± 0.001400	First order	0.9755	99.99

## Data Availability

The data presented in this study are available on request from the corresponding author.

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
