# Peer review of "Development of a Hydrophobicity-Controlled Delivery System Containing Levodopa Methyl Ester Hydrochloride Loaded into a Mesoporous Silica"

_pharmaceutics, 2021, doi:10.3390/pharmaceutics13071039_

Round 1

Reviewer 1 Report

In this paper, Ambrus and colleagues reported a study on drug release control by loaded hydrophilic hydrophobized mesoporous silica particles (MSP). The study was complete, starting from the modification of commercial MPS, followed by loading studies and experiment on the release in the function of different parameters. 
MPS are very popular since few years in the biomedical fields, essentially because it is possible to easily engineering them and because they are biologically compatible, so a lot of studies were performed targeting different diseases and even diagnostic applications.
In this case, the target is Parkinson, one of the most difficult fields of medicine so, even small advancement in the knowledge and applications can be of huge importance.
I think that the study presented is not revolutionary, but it is very well performed, by using a rich plethora of techniques for the different characterisations handled. The result will bring interesting food for thoughts and hints for the researchers working in the field by providing useful data (as an example, it is interesting the capacity to push or not the crystallisation of the API, once loaded, but also others); for these reasons, I suggest accepting the paper for publications on Pharmaceutics.
I have a suggestion to the authors because in my field (nanotech) it would not be considered the role of MPS as an “excipient” role, but much more as a vector that can be considered “active”; it is because of the characteristic (chemical and physical) of the MSP and because of the engineering of these particles that some enhanced results were obtained, so the “secondary” role assigned to MSP seems to me quite reductive.

Author Response

Dear Reviewer!

Thank you for your opinion, we hope that numerous researchers will be interested in this work, utilize our results and get a good inspiration for their own work.

Thank you for your suggestion. We can accept your point of view, avoiding the use of word ”excipient” for MPS as they are applied in different research area and it also has an active influence on the properties of product, therefore we changed the ”excipient” word throughout the manuscipt to ”carrier”, which might describe the role of MPSs more precisely. The corrections related to your suggestion are highlighted with yellow colour.

Szeged, 30th June 2021

Tamás Kiss, Rita Ambrus

Reviewer 2 Report

The study about the controlling release of LDME by hydrophobicity of silica supports is interesting and important for the drug formulation, but the  manuscript could be improved .

The references provided are too general and do not emphasize the release control of hydrophilic drugs . A good example is the paper Controlling drug release from mesoporous silica through an amorphous, nanoconfined 1-tetradecanol layer (R.-A. Mitran et al.) European Journal of Pharmaceutics and Biopharmaceutics, 127 (2018) 318-325.

The drug concentration limits in patient body are not discussed.

Instead of the mentioned Hungarian companies that are distributors, give the developing companies.

The text between lines 94-101 is fuzzy. Generally speaking entire manuscript should be polish.

 Fig 5. caption should be revised. Also, modify/add Intensity (a.u.) instead of Relative Intensity, in DSC graphs use Energy and similar scale for C and D.

Chapter 2.16 refers to LDME stability in aqueous medium and not to the drug release

-CH3 vibration bands are at 2850 and 2950 cm-1.

Table 5 wLDME (g/g)

Conclusions should be focused on results and their correlation and less on describing the working steps.

Author Response

Dear Reviewer,

thank you for reviewing our article. The changes in connection with your comments are highlighted with red colour in the article.

The references provided are too general and do not emphasize the release control of hydrophilic drugs . A good example is the paper Controlling drug release from mesoporous silica through an amorphous, nanoconfined 1-tetradecanol layer (R.-A. Mitran et al.) European Journal of Pharmaceutics and Biopharmaceutics, 127 (2018) 318-325.

Thank you for this comment. We have completed the manuscript with more suitable references which dealed with control of hydrophillic drugs according to the suggestion. These references are the following:

  1. Mitran, R.A.; Matei, C.; Berger, D.; Băjenaru, L.; Moisescu, M.G. Controlling drug release from mesoporous silica through an amorphous, nanoconfined 1-tetradecanol layer. European Journal of Pharmaceutics and Biopharmaceutics 2018, 127, 318-325, doi:10.1016/j.ejpb.2018.02.020.
  2. Chen, X.; Liu, Z. Dual responsive mesoporous silica nanoparticles for targeted co-delivery of hydrophobic and hydrophilic anticancer drugs to tumor cells. Journal of Materials Chemistry B 2016, 4, 4382-4388, doi:10.1039/C6TB00694A.
  3. Liu, Q.; Zhang, J.; Sun, W.; Xie, Q. R.; Xia, W.; Gu, H. Delivering hydrophilic and hydrophobic chemotherapeutics simultaneously by magnetic mesoporous silica nanoparticles to inhibit cancer cells. International journal of nanomedicine 2012, 7, 999, doi:10.2147/IJN.S28088.

The drug concentration limits in patient body are not discussed.

The literature data about LDME is really limited, there is no information about the concentration limit. Stocchi et al details that the blood level of LDME from LDME/carbidopa effervescent tablet in the comparsion of standard-release levodopa/carbidopa tablets. The cmax was in the range of 1000-3500 ng/ml. In this range, no adverse effect was detected. (Reference: Stocchi, F.; Vacca, L.; Grassini, P.; Pawsey, S.; Whale, H.; Marconi, S.; Torti, M. L-dopa pharmacokinetic profile with effervescent melevodopa/carbidopa versus standard-release levodopa/carbidopa tablets in Parkinson’s disease: a randomised study. Parkinson’s Disease 2015, 2015, doi:10.1155/2015/369465.)

The doses of commercially available LDME containing effervescent tablets is 314 mg of LDME/25 mg of carbidopa; 157 mg of LDME/12.5 mg of carbidopa; 125.6 mg of LDME/25 mg of carbidopa.

Instead of the mentioned Hungarian companies that are distributors, give the developing companies.

The chapter ”2.1. Materials” was changed based on your suggestions. Thank you for them. Now, the manufacturer is indicated in the text with the international headquarter in the bracket.

The text between lines 94-101 is fuzzy. Generally speaking entire manuscript should be polish.

Thank you for this comment!

We tried to correct the errors that made the text fuzzy. The text between lines 91-101 was changed to following version:

”As the conventional theoretical methods struggle to deal with drug release kinetics, the development of a universal approach would be vital to this field. A promising but not yet popular approach utilizes a so-called lumped second-order kinetic model to address mixed release kinetics [30]. However, to the best of our knowledge, it has not been tested how efficiently this equation can describe the release kinetics of different, widely used re-lease models. Its application to the release from MPSs has not been investigated, either.”

Fig 5. caption should be revised. Also, modify/add Intensity (a.u.) instead of Relative Intensity, in DSC graphs use Energy and similar scale for C and D.

Thank you for this comment.

Based on you proposal, we changed the ordinate axis of XRPD diffractograms to ”Intensity (a.u.)” from ”Relative intensity” and added the name of ordinate axis of DSC curve which was ”Normalized heat flow” which was not on the figure, indeed. We think it is more accurate to call it normalized heat flow because if it was energy, the unit should be joule. After integrating the curve, the result would be joule, indeed. You can see the corrected Figure 5 here:

Chapter 2.16 refers to LDME stability in aqueous medium and not to the drug release

You are right, thank you for this remark!

We renamed the Chapter 2.16. to ”Investigation of degradation kinetics of LDME”.

-CH3 vibration bands are at 2850 and 2950 cm-1.

Thank you for this comment, as well!

Thanks for it, we completed the maniscript with the following text:

”Besides, new peaks appear at 2906 cm-1 and 2964 cm-1 which are corresponding to the C-H vibration of the trimethylsilyl groups.”

This issue was noticed by Reviewer 3, too. This is the reason why this sentence is blue and red at the same time.

Table 5 wLDME (g/g)

Thank you for the remark, we have completed the title of this column.

Conclusions should be focused on results and their correlation and less on describing the working steps.

Thank you for this remark, too!

We added new parts to the Conclusions which are highlighted with red colour. I hope you find the changes satisfying.

The new version of Conclusions is also copied here:

”In this work, the drug release control from loaded hydrophilic and hydrophobized MPSs was investigated. MPS was hydrophobized by TMCS, the model API was LDME. To analyze the release properties of hydrophobized and non-hydrophobized MPS, a 32 full-factorial design was set and executed. The independent variables were the TMCS concentration (0, 10, 20 mM) and the LDME mass percent at the sample preparation (5, 12.5 and 20 % w/w). The wettability of MPSs was reduced by using TMCS, the silanol group density decreased and the trimethylsilyl groups were attached to the surface. LDME was loaded into the mesopores by solvent drying method using rotavapor. The API was homogenously distributed, secondary interactions were developed between the surface and the drug. The surface-drug interaction seemed to be stronger when the SYL was more hydrophilic presumably causing that the LDME was sporadically in the case of SYL-1-5 and SYL-2-5 compared to SYL-0-5. The drug release from the powder could be significantly decreased by the hydrophobization of MPS verified by the evaluation of ANOVA and the Tukey HSD tests. The release rate could be significantly decreased in every product compared to the LDME reference powder, except for SYL-0-20. The release rate was evaluated by fitting to kinetic models and parameter a, which characterized the IRR. The adaptability of this model was also verified for different release kinetics with the help of the similarity factor which was 99.91 or higher showing that it can be used in the case of different release kinetics but the similarity factor is suggested to check between the actual release values and the points of model curve. The LDME release significantly decreased due to hydrophobization (p<0.05) in the 5-20 % w/w LDME range. However, the physical stability of the products was not sufficient when the API mass percent was 20 % w/w. In the case of the SYL-0-20 product, a weak sign of crystalline fraction appeared in the X-ray diffractogram after 3 months of storage at -20 °C, while the API was not chemically stable at 40 °C, 75 % RH. Higher amounts of LDME increased the value of IRR, i.e. drug release, however, the physical stability of the material loaded into the mesopores limits scientists and developers in preparing MPS containing products with a high API content.

The use of hydrophobized MPSs can be recommended for APIs with a narrow therapeutic index because the API release can be systematically regulated by controlling the wettability of MPSs and the API content. The optimized formulations with zero order release kinetics can be ideal for preparing a formulation that can provide steady blood levels resulting in shortening of off periods, i.e. reducing the side effects compared to traditional per os levodopa treatment. The same approach is used in the case of LD/carbidopa containing duodenal infusion [48].”

Szeged, 30th June 2021

Tamás Kiss, Rita Ambrus

Reviewer 3 Report

The manuscript submitted by Professor Rita Ambrus reports a strategy for preparation Syloid® based drug delivery systems for controlled release of melevodopa hydrochloride (LDME) via silylation the Syloid® 103 XDP 3050 with trimethylchlorosilane (TMCS). The properties of the hydrophobized Syloid® as well as materials loaded with LDME in relation to porosity, distribution of LDME, the physical stability and the control of release of loaded molecules are comprehensively discussed. In my opinion the topic is very interesting and the presented results are sound. The article contains precise description and a comprehensive discussion of the presented data. The results and the discussion are supported by the appropriate literature. The scientific content is sound. Therefore, I recommend publication of the manuscript as it is. However, the authors did not avoid a few errors and inconsistencies. They should be clarified before the publication of the article.

# line 24 - An acronym “LDME” should appear on line 22, right after the full name of the drug.

# line 236 - The commonly used acronym for X-ray powder diffractometry is XRD. XRPD is used less frequently, but XRP is rarely seen and is rather confusing. I strongly recommend to use the XRD shortcut.

# line 280 - Why was the phosphate buffer with pH 6.8 containing NaOH and KH2PO4 used instead of buffer with pH 6.8 recommended by the European Pharmacopoeia?

# line 283 - A sink condition should be maintained during the dissolution experiment, according to the pharmacopeia requirements for capsules. 100 ml of the release medium for 90 mg of the investigated system seems to be not enough. Moreover, the portion of 3 ml of sample (taken at predetermined time intervals) seems to be too much.

# FTIR data - The presence of the absorption bands around 3000 cm-1 in FTIR spectra of SYL-1 and SYL-2 should be discussed since they are characteristic for the organic moieties anchored to the silica surface.

Author Response

Dear Reviewer!

Thank you for the kind words, based on you suggestion, we corrected these errors. The corrected version in the manuscript is highlighted with blue colour.

# line 24 - An acronym “LDME” should appear on line 22, right after the full name of the drug.

Thank you for this remark!

We have added the acronym ”LDME” to the content of bracket, besides it was completed with the another name of this API, i.e. levodopa methyl ester hydrochloride.

# line 236 - The commonly used acronym for X-ray powder diffractometry is XRD. XRPD is used less frequently, but XRP is rarely seen and is rather confusing. I strongly recommend to use the XRD shortcut.

Thank you for you comment!

We intended to emphasize that we applied X-ray powder diffractometry or powder X-ray diffractometry and not single-crystal X-ray diffractometry, therefore we used the term XRPD as the XRD could refer to both. Thus, it is more accurate to use XRPD instead of XRD. The ”XRP diffractogram” expression is actually rarely used, so it might really look improper, therefore it was changed to ”diffractogram” for the sake of disambiguation.

# line 280 - Why was the phosphate buffer with pH 6.8 containing NaOH and KH2PO4 used instead of buffer with pH 6.8 recommended by the European Pharmacopoeia?

The pH=6.8 phosphate buffer was prepared based on the chapter of ”5.17.1. Recommendations on dissolution testing” of Eur. Ph. 8.4. was used as a basis. We used the buffer solution that can be used for simulated intestinal fluid pH 6.8 after adding pancreas powder R to this. The effect of pH was really important for us because mainly it affects the stability of SYLs and LDME at 37 °C.

# line 283 - A sink condition should be maintained during the dissolution experiment, according to the pharmacopeia requirements for capsules. 100 ml of the release medium for 90 mg of the investigated system seems to be not enough. Moreover, the portion of 3 ml of sample (taken at predetermined time intervals) seems to be too much.

Thank you for this comment!

The solubility of LDME is really high, therefore the sink condition was achieved. The solubility of LDME is 912 mg/ml which was measured by our research group and detailed in an article, named ”Crystallization and physicochemical investigation of melevodopa hydrochloride, a commercially available antiparkinsonian active substance”, accepted for publication in Acta Pharmaceutica Hungarica. In this article, the products contained maximum 19.26 mg of LDME, so even very small volume (0.211 ml) of dissolution media should have been enough to satisfy the sink condition. The 3 ml volume of samples was chosen because the results were also obtained by spectrophotometrically in the first rounds but then the HPLC quantification seemed to be more accurate. Because of the spectrophotometry, we needed to have higher amount of samples, after that we intended to sample the same amounts to have parallels.

# FTIR data - The presence of the absorption bands around 3000 cm-1 in FTIR spectra of SYL-1 and SYL-2 should be discussed since they are characteristic for the organic moieties anchored to the silica surface.

Thank you for this comment, as well!

We completed the maniscript with the following text:

”Besides, new peaks appear at 2906 cm-1 and 2964 cm-1 which are corresponding to the C-H vibration of the trimethylsilyl groups.”

This issue was noticed by Reviewer 2, too. This is the reason why this sentence is blue and red at the same time.

Szeged, 30th June 2021

Tamás Kiss, Rita Ambrus
